# Structure-guided design of a selective inhibitor of the methyltransferase KMT9 with cellular activity

Sheng Wang [1], Sebastian O. Klein[2], Sylvia Urban[1], Maximilian Staudt [3], Nicolas P. F. Barthes [3], Dominica Willmann[1], Johannes Bacher[3], Manuela Sum[1], Helena Bauer[1], Ling Peng[1], Georg A. Rennar[3], Christian Gratzke[1], Katrin M. Schüle[4], Lin Zhang [5], Oliver Einsle[5], Holger Greschik[1], Calum MacLeod[6], Christopher G. Thomson[6], Manfred Jung [2,3,7], Eric Metzger [1,7] ✉ & Roland Schüle [1,2,7] ✉

Inhibition of epigenetic regulators by small molecules is an attractive strategy for cancer treatment. Recently, we characterised the role of lysine methyltransferase 9 (KMT9) in prostate, lung, and colon cancer. Our observation that the enzymatic activity was required for tumour cell proliferation identified KMT9 as a potential therapeutic target. Here, we report the development of a potent and selective KMT9 inhibitor (compound 4, KMI169) with cellular activity through structure-based drug design. KMI169 functions as a bisubstrate inhibitor targeting the SAM and substrate binding pockets of KMT9 and exhibits high potency, selectivity, and cellular target engagement. KMT9 inhibition selectively downregulates target genes involved in cell cycle regulation and impairs proliferation of tumours cells including castration- and enzalutamide-resistant prostate cancer cells. KMI169 represents a valuable tool to probe cellular KMT9 functions and paves the way for the development of clinical candidate inhibitors as therapeutic options to treat malignancies such as therapy-resistant prostate cancer.

Protein methyltransferases (PMTs) are extensively explored as therapeutic targets[1,2] since abnormal expression or genomic alterations of these proteins result in various diseases including cancer[3–5]. PMTs catalyse the transfer of methyl groups from the cofactor S-5'-adenosyl-L-methionine (SAM) to either lysine or arginine residues of histones and non-histone proteins[6,7]. Based on the structure of the catalytic domain, PMTs are classified as either Su(var)3-9, enhancer of zeste (E(z)), and trithorax (trx)[8] (SET) domain- or Rossmann fold-containing family members. Histone lysine methylation was thought to be generally carried out by PMTs of the SET domain family[2] whereas

Rossmann fold family members were primarily observed to act as RNA or protein arginine methyltransferases[9].

Recently, we identified the Rossmann-fold family member KMT9 as a histone lysine methyltransferase monomethylating histone H4 at lysine 12 (H4K12me1)[10]. KMT9 is an obligate heterodimer composed of KMT9α (also known as N6AMT1) and KMT9β (also known as TRMT112). KMT9α contains a typical Rossmann-fold SAM binding pocket. KMT9β acts as a chaperone protein and the interaction of both subunits is required for SAM binding and methyltransferase activity. KMT9 controls the growth of several types of tumour cells including prostate,

[1]Klinik für Urologie und Zentrale Klinische Forschung, Klinikum der Albert-Ludwigs-Universität Freiburg, Freiburg, Germany. [2]CIBSS Centre of Biological Signalling Studies, University of Freiburg, Freiburg, Germany. [3]Institute of Pharmaceutical Sciences, Albert-Ludwigs-Universität Freiburg, Freiburg, Germany. [4]Institute of Experimental and Clinical Pharmacology and Toxicology, Faculty of Medicine, University of Freiburg, Freiburg, Germany. [5]Institut für Biochemie, Albert-Ludwigs-Universität Freiburg, Freiburg, Germany. [6]Drug Discovery, Pharmaron UK Ltd, Hoddesdon, United Kingdom. [7]Deutsches Konsortium für Translationale Krebsforschung, Standort Freiburg, Freiburg, Germany. ✉e-mail: eric.metzger@uniklinik-freiburg.de; roland.schuele@uniklinik-freiburg.de

lung, and colon cancer cell lines[10–12]. KMT9 was also reported to methylate glutamine residues in the eukaryotic translation termination factor 1 in vitro[13–15] or to act as an adenine-N6 methyltransferase, an observation that remains controversial[16–18]. KMT9 depletion blocks proliferation of androgen receptor-dependent as well as castration- and enzalutamide-resistant prostate cancer cells via deregulation of gene expression and induction of apoptosis[10]. In comparison, non-apoptotic cell death was observed to account for the anti-proliferative effect of KMT9 depletion in small cell and non-small cell lung cancer cells[11]. Furthermore, KMT9 was shown to regulate colorectal tumor-igenesis by affecting maintenance and function of colorectal cancer stem cells[12]. Of note, using a catalytically inactive mutant, we provided evidence that tumour cell proliferation is controlled by enzymatically active KMT9[10]. Therefore, selective inhibition of the catalytic activity of KMT9 with a small-molecule inhibitor offers as a potential therapeutic approach to impair KMT9-dependent cancer cell proliferation.

Here, we report the development of a potent and selective small-molecule KMT9 inhibitor (KMI169). KMI169 was optimized through structure-based drug design and displays high selectivity for KMT9 against a large panel of SET domain- or Rossmann fold-containing methyltransferases as well as protein kinases. The inhibitor exhibits nanomolar potency in vitro and cellular KMT9 target engagement. Inhibition of KMT9 catalytic activity results in reduced cellular levels of H4K12me1, down-regulation of KMT9 target genes involved in cell cycle regulation and, in consequence, in the suppression of prostate tumour cell growth. Taken together, we identified KMT9 as a drug-gable target and present KMI169 as an inhibitor with cellular activity to unravel KMT9 biology. Since KMI169 severely impairs proliferation of castration- and enzalutamide-resistant prostate cancer cells, the compound is a promising starting point for the development of clinical candidate KMT9 inhibitors empowering novel therapeutic strategies for the treatment of therapy-resistant prostate cancer.

## Results

### Identification of a KMT9 bi-substrate inhibitor

To initiate the development of KMT9 inhibitors, we first addressed the question whether it was feasible to follow a 'bi-substrate inhibitor' strategy[1,19,20]. For this purpose and based on our previously reported crystal structure of the KMT9/S-adenosyl-L-homocysteine (SAH)/H4K12me1 peptide complex (PDB code 6H1E[10]), we linked the SAH backbone with a mimic of the lysine side chain in the substrate channel by replacing the sulphur with a nitrogen atom resulting in compound 1a (Fig. 1a, Supplementary Fig. 1a). Compound 1a consists of an ade-nosine moiety, an 'amino acid branch', and a 'substrate branch' pre-dicted to bind into the KMT9α adenosine pocket, the methionine pocket, and the substrate channel, respectively (Fig. 1a).

Using Microscale Thermophoresis (MST), we determined a dis-sociation constant ($K_d$) of 0.006 μM for KMT9 binding of compound 1a (Fig. 1b). Consistently, in Fluorescent Thermal Shift Assay (FTSA), we observed a strong melting temperature shift ($\Delta T_m$) of 17.0 K relative to dimethylsulfoxide (DMSO) control (Fig. 1c). In comparison, for S-adenosyl-L-methionine (SAM) and SAH we determined $K_d$ values of 2.9 μM and 10.9 μM and ΔTm values of 6.8 K and 6.9 K, respectively (Fig. 1b, d, e and Supplementary Fig. 1b-d). Thus, our bi-substrate approach resulted in a high affinity KMT9 inhibitor.

We next solved the crystal structure of the KMT9/compound 1a complex. Superimposition with the KMT9/SAH/H4K12me1 peptide structure[10] confirmed identical binding modes for the SAH backbones of the ligands (Supplementary Fig. 1e). As expected, the amino acid and the substrate branches of compound 1a occupied their respective binding pockets (Fig. 1f). Of note, the amino group in the substrate channel formed hydrogen bonds with the side chains of D28 and N122 as well as the main chain carbonyl group of P123 (Fig. 1g) and was involved in a water-mediated hydrogen bonding network comprising Y23, Y125, W142, and A141 of KMT9α (Supplementary Fig. 1f, g). The

differences in KMT9 binding between compound 1a and SAH sug-gested that the intensive bonding network formed by the 'amine anchor' in the substrate channel accounted for the high affinity of compound 1a. We reasoned that comparable protein-ligand interac-tions in the substrate channel would be a prerequisite to achieve high potency with alternative ligand designs.

To explore whether the KMT9/compound 1a crystal structure could serve as a reliable template for the modelling of new ligands, we designed, characterised, and co-crystallised several compound 1a derivatives exemplified by compounds 1b and 1c (Supplementary Fig. 2a-d). The KMT9/compound 1a, 1b, and 1c crystal structures revealed a good superimposition of the ligands and only marginal conformational changes in the SAM binding pocket and the substrate channel of KMT9α (Supplementary Fig. 2e, f). Furthermore, differ-ences in ligand binding affinity could be explained, at least in part, by a reduced number of hydrogen bonds formed by compounds 1b and 1c compared to compound 1a in the substrate channel. Together, our observations suggested an overall high rigidity of the bi-substrate binding pocket allowing modelling approaches for analogues of compound 1a.

### Modification of the ligand scaffold to improve biophysical properties

SAM-derived ligands often do not possess favourable drug-like prop-erties resulting, for example, in low cell membrane permeability due to the polar SAM backbone[19]. To improve compound 1a, we followed several routes. On the one hand, we aimed to reduce compound flexibility and charge by substituting the amino acid and substrate branches. In addition, we focused on optimising biophysical com-pound properties [calculated lipophilicity (clogD) and Topological Polar Surface Area (TPSA)], for example, by replacement of oxygen and basic nitrogen with carbon atoms. Accordingly, we modelled and tes-ted alternative scaffolds (compounds 2a-e; Supplementary Fig. 3a) and finally decided to replace the hydrophilic amino acid and substrate branches of compound 1a with a series of substituted phenyl moieties (compounds 2a and 2f-h; Fig. 2a).

The substitutions (-H, -$NH_2$, -$CH_2$-$NH_2$, -$(CH_2)_2$-$NH_2$) explored the potential to include an amine anchor in the new scaffold to maintain high potency. In addition, in all series 2 compounds (Fig. 2a, Supple-mentary Fig. 3a), the 7-nitrogen of adenine and the 5'-oxygen of ribose were replaced with carbon atoms since analysis of the KMT9/com-pound 1a structure showed that these atoms were embedded in a mostly hydrophobic protein environment (Supplementary Fig. 3b). In comparison, in SET domain-containing methyltransferases, the 7-nitrogen atom is buried in a small pocket forming a specific hydro-gen bond with the protein backbone[2,21] (Supplementary Fig. 3c). Therefore, replacement of the 7-nitrogen with carbon may improve the selectivity versus SET domain-containing methyltransferases. Throughout the manuscript, we refer to the modified adenosine moiety as 'adenosine mimic'. Compared to compound 1a, these modifications strongly improved the calculated biophysical properties of the molecules (compounds 2a and 2f-h; Fig. 2a). The most pro-mising compound in this series (compound 2 g; clogD = −1.8, TPSA = 126.1) carried a methylamine anchor attached to the phenyl moiety and exhibited a $K_d$ of 1 μM (Fig. 2a). A shorter (compound 2 f) or longer (compound 2 h) amine anchor decreased KMT9 binding (Fig. 2a).

To improve the binding affinity of compound 2 g, we next aimed to identify and fill unoccupied KMT9 sub-pockets. In our KMT9/com-pound 1a crystal structure we identified such an unoccupied space adjacent to the 2-position of the adenine (sub-pocket A), which was formed by polar and hydrophobic residues of KMT9α (Supplementary Fig. 3d). Modelling suggested that a chlorine atom would perfectly fill sub-pocket A and potentially form both hydrophobic interactions with surrounding residues and a halogen bond with the side chain of T76 (Fig. 2d). Indeed, addition of a chlorine atom at the 2-position of

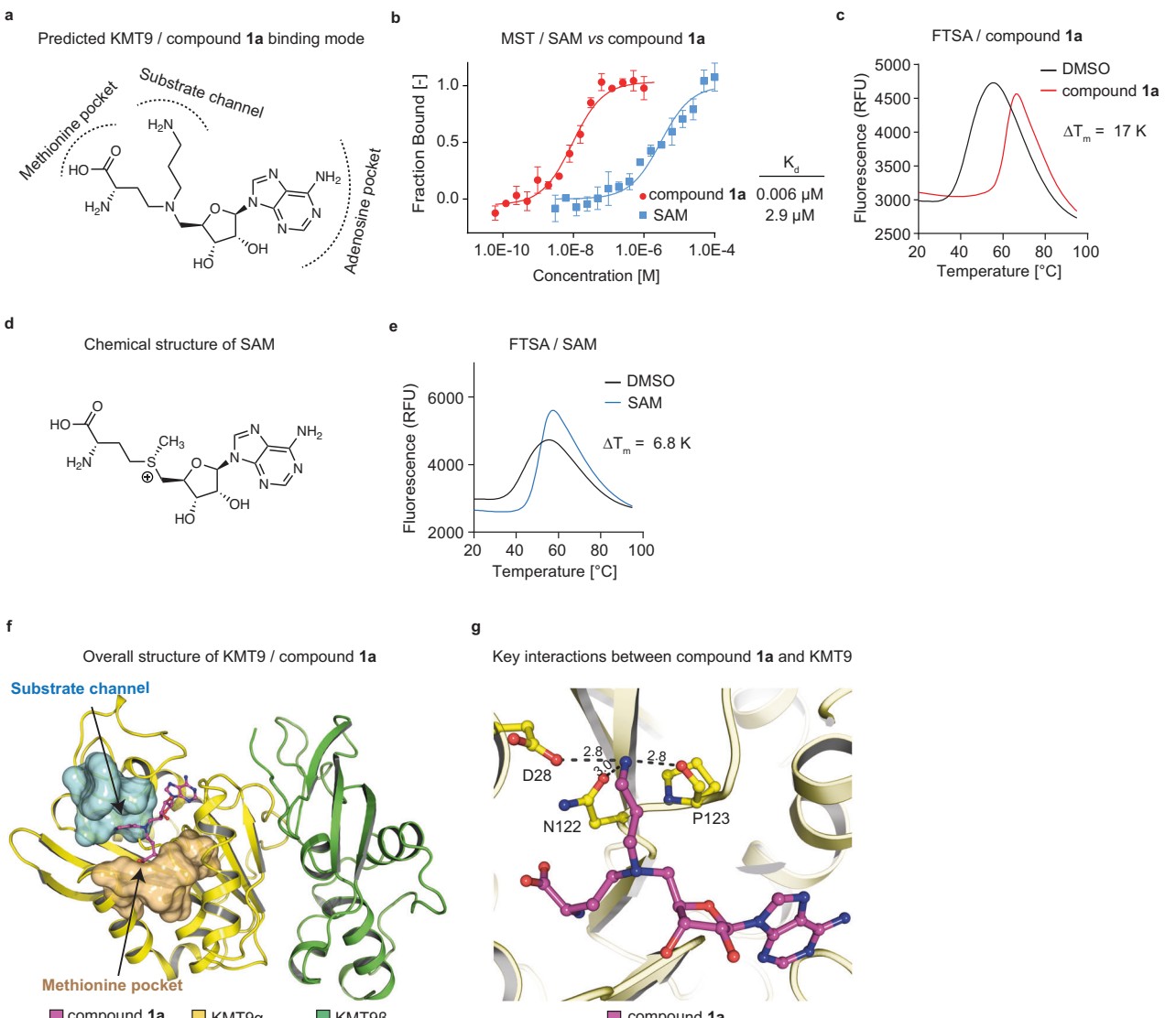

**Fig. 1 | Identification of a bi-substrate inhibitor of KMT9. a** Chemical structure of compound **1a** with predicted binding mode. **b** MST assay to determine the dissociation constant ($K_d$) of compound **1a** and SAM binding to KMT9. Data represent means ±s.d ($n = 3$ independent experiments). **c** FTSA for compound **1a** binding to KMT9. $\Delta T_m$ was calculated relative to vehicle (DMSO). Data represent means ($n = 4$). **d** Chemical structure of SAM. **e** FTSA for SAM binding to KMT9. Data represent means ($n = 4$). **f** Overall structure of the KMT9/compound **1a** complex (PDB code

8CNC). KMT9α (yellow) and KMT9β (green) proteins are represented as ribbons. Arrows indicate the substrate channel (cyan) and the methionine pocket (brown) illustrated by surface view. Compound **1a** is shown as sticks (magenta). **g** Hydrogen bonds between the amino group of compound **1a** (magenta) and KMT9α (yellow) in the substrate branch. KMT9α is shown as ribbon. Key residues and ligands are depicted as sticks. Contacts are represented by black dashed lines.

compound **2 g** (resulting in compound **3a**) strongly increased the binding affinity by about 100-fold ($K_d = 0.008\,\mu M$; Fig. 2b). To attenuate the polarity and reduce the number of hydrogen bond donors of compound **3a**, we replaced the free amine with an azetidine moiety (compound **3b**; Fig. 2c). $K_d$ and $\Delta T_m$ of compound **3b** ($K_d = 0.015\,\mu M$, $\Delta T_m = 9.1\,K$) were only slightly reduced compared to compound **3a** ($K_d = 0.008\,\mu M$, $\Delta T_m = 10.7\,K$). On the other hand, compound **3b** exhibited improved biophysical properties (clogD = 1.0, TPSA = 101.6) compared to compound **3a**. Modelling suggested that the azetidine moiety forms charged interactions with the side chain of D28 as well as favourable cation-π interactions with the side chains of Y23 and Y125 (Fig. 2e). To evaluate the potency of compounds **3a** and **3b** to inhibit KMT9, we performed enzymatic inhibition assays of KMT9 in vitro, which revealed $IC_{50}$ values of $0.12\,\mu M$ and $0.20\,\mu M$, respectively (Fig. 2f, g). Together, compounds **3a** and **3b** exhibited binding affinities in the low nanomolar range, inhibited the catalytic activity of KMT9, and possessed favourable biophysical properties.

To achieve additional improvements with respect to potency and cell membrane permeability, we further modified the adenosine mimic of compound **3b**. In the KMT9/compound **1a** crystal structure, we identified an additional space filled with water molecules close to the 7-nitrogen of the adenosine moiety (sub-pocket B; Supplementary Fig. 3e). Docking suggested that a larger substituent would fit into sub-pocket B of KMT9 after manually removing the water molecules. Based on our modelling, a 2-chloro-pyrazole was eventually selected (Fig. 2h). Accordingly, the chlorine atom of the 2-chloro-pyrazole may enable hydrophobic or halogen-mediated interactions with P124, V133, and A143 of KMT9α (Fig. 2h). In addition, rotation of the pyrazole moiety might be hindered by an intramolecular hydrogen bond between pyrazole and the 6-amino group of the adenosine mimic resulting in reduced compound flexibility, which should be beneficial for potency and cell membrane permeation. The 2-chloro-pyrazole could be further extended (Fig. 2i) by adding a benzyl moiety to the 4-nitrogen atom resulting in

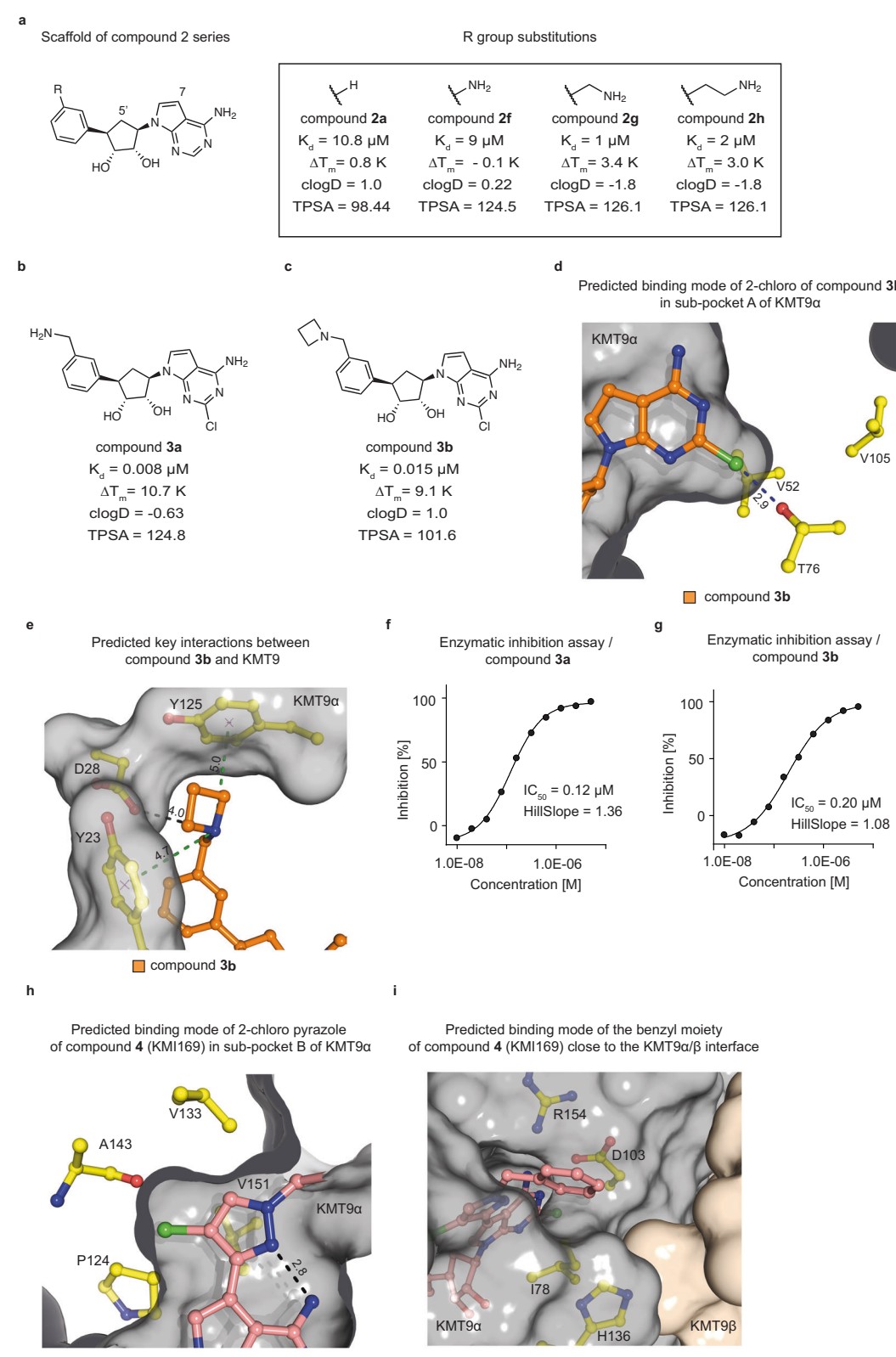

**a** Scaffold of compound 2 series

R group substitutions

compound **2a**
$K_d$ = 10.8 µM
$\Delta T_m$ = 0.8 K
clogD = 1.0
TPSA = 98.44

compound **2f**
$K_d$ = 9 µM
$\Delta T_m$ = - 0.1 K
clogD = 0.22
TPSA = 124.5

compound **2g**
$K_d$ = 1 µM
$\Delta T_m$ = 3.4 K
clogD = -1.8
TPSA = 126.1

compound **2h**
$K_d$ = 2 µM
$\Delta T_m$ = 3.0 K
clogD = -1.8
TPSA = 126.1

**b**

compound **3a**
$K_d$ = 0.008 µM
$\Delta T_m$ = 10.7 K
clogD = -0.63
TPSA = 124.8

**c**

compound **3b**
$K_d$ = 0.015 µM
$\Delta T_m$ = 9.1 K
clogD = 1.0
TPSA = 101.6

**d** Predicted binding mode of 2-chloro of compound **3b** in sub-pocket A of KMT9α

KMT9α
V105
V52
T76
2.9
■ compound **3b**

**e** Predicted key interactions between compound **3b** and KMT9

Y125  KMT9α
D28
Y23
5.0
4.0
4.1
■ compound **3b**

**f** Enzymatic inhibition assay / compound **3a**

Inhibition [%]
$IC_{50}$ = 0.12 µM
HillSlope = 1.36
Concentration [M]

**g** Enzymatic inhibition assay / compound **3b**

Inhibition [%]
$IC_{50}$ = 0.20 µM
HillSlope = 1.08
Concentration [M]

**h** Predicted binding mode of 2-chloro pyrazole of compound **4** (KMI169) in sub-pocket B of KMT9α

V133
A143
V151
KMT9α
P124
2.8
■ compound **4** (KMI169)

**i** Predicted binding mode of the benzyl moiety of compound **4** (KMI169) close to the KMT9α/β interface

R154
D103
I78
KMT9α
H136
KMT9β
■ compound **4** (KMI169)

compound **4**, hereafter named KMI169 (Fig. 3a). The added benzyl moiety may bind to a surface formed by the side chains of I78, D103, and H136 close to the KMT9α/β interface (Fig. 2i). We also designed an inactive control compound (KMI169Ctrl) by deletion of the 2-chloro and 6-amino substituents and addition of two fluorine atoms to the azetidine moiety to neutralize its basicity (Fig. 3b). KMT9 binding of KMI169 ($K_d$ = 0.025 µM; Fig. 3c) was slightly decreased compared to compound **3b**, while the thermal stabilization ($\Delta T_m$ = 12.5 K; Fig. 3d) and KMT9 inhibition were improved ($IC_{50}$ = 0.05 µM; Fig. 3e). In contrast, KMI169Ctrl neither bound to KMT9 nor showed thermal stabilization or affected its enzymatic activity (Fig. 3c-e). Next, we measured the $IC_{50}$ values for inhibitor binding in the presence of varying concentrations of SAM or substrate (Fig. 3f, g). KMI169 $IC_{50}$ values displayed a linear dependence

**Fig. 2 | Structure-guided lead optimization of KMT9 inhibitors. a** Chemical structures of the compound **2** series. The scaffold structure (left) and the R group substitutions (right) for compounds **2a** and **2f-h** are represented. Measured $K_d$ and $\Delta T_m$ values and calculated clogD and TPSA values for each compound are listed. **b, c** Chemical structures of compounds **3a** and **3b**. Measured $K_d$ and $\Delta T_m$ values and estimated clogD and TPSA values are listed. **d, e** Predicted binding of the 2-chloro substituent of compound **3b** (orange) into sub-pocket A of KMT9α (**d**) and key interactions between compound **3b** and KMT9α residues (yellow) (**e**). KMT9α is shown as grey surface. Key residues and ligand are shown as sticks. Contacts are represented by blue (halogen bond) (**d**) green (cation-π interactions); (**e**) or black dashed lines (charged interaction) (**e**). **f, g** Enzymatic inhibition of KMT9 by compounds **3a** (**f**) and **3b** (**g**). Data represent means (*n* = 2 independent experiments). **h, i** Representations of the predicted binding modes of the 2-chloro-pyrazole moiety of compound **4** (KMI169) (pink) in sub-pocket B of KMT9α (**h**) and of the benzyl substituent of compound **4** (KMI169) at the KMT9α/β interface (**i**). KMT9α (grey) and KMT9β (brown) are shown as surface view. KMT9α key residues (yellow) and ligands are shown as sticks. The intra-molecular hydrogen bond in the 2-chloro-pyrazole moiety (**h**) is represented by a black dashed line.

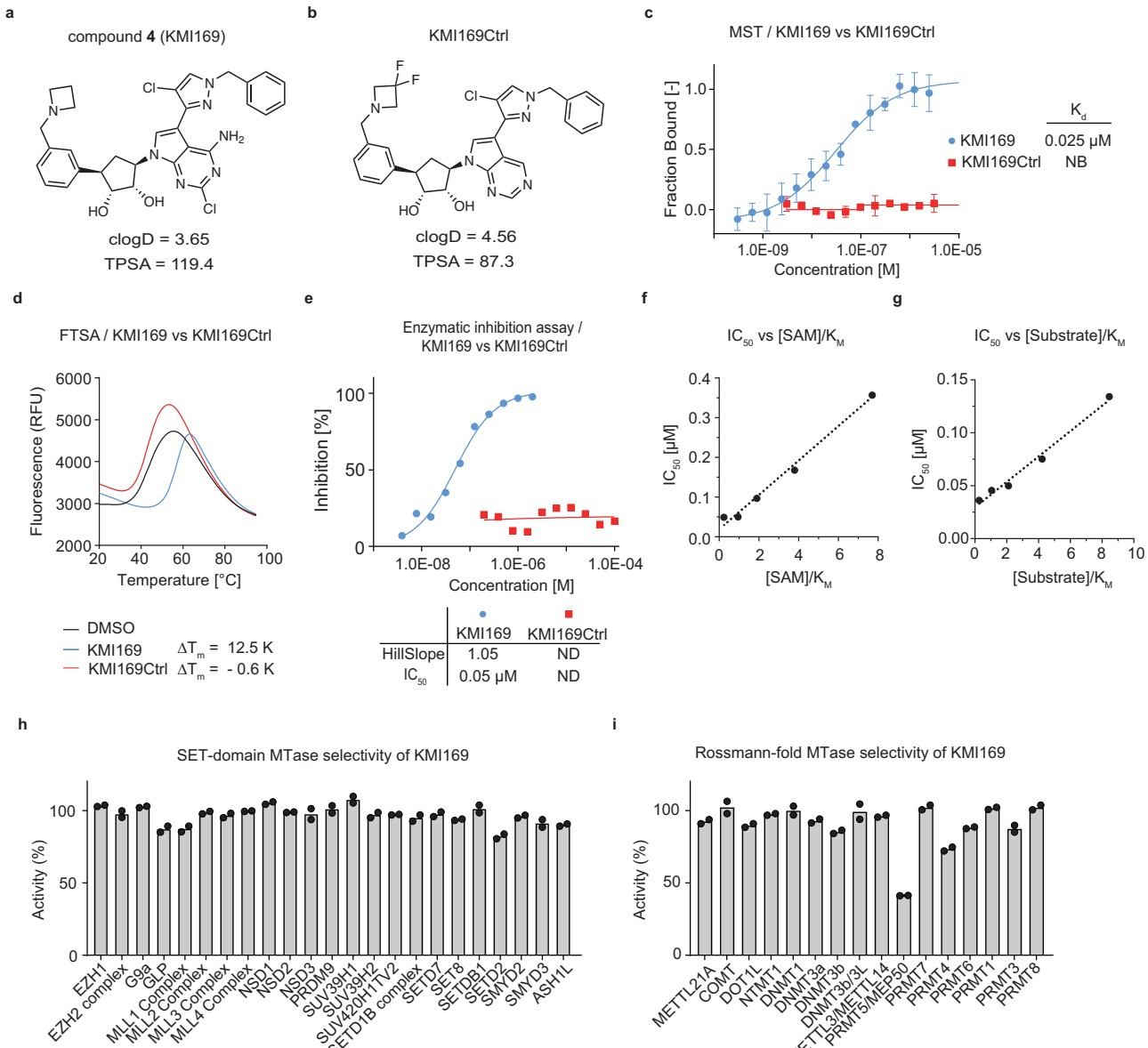

**Fig. 3 | Characterization of compound 4 (KMI169). a** Chemical structures of compound **4** (KMI169) (**a**) and KMI169Ctrl (**b**). **c** MST assay to determine the $K_d$ of KMI169 and KMI169Ctrl binding to KMT9. NB: no binding. Data represent means ± s.d (*n* = 3 independent experiments). **d** FTSA for KMI169 and KMI169Ctrl binding to KMT9. Data represent means (*n* = 4 independent experiments). **e** Enzymatic inhibition of KMT9 by KMI169 and KMI169Ctrl. Data represent means (*n* = 2 independent experiments). **f, g** IC$_{50}$ values for inhibition of KMT9 by KMI169 plotted as function of [SAM]/K$_M$ (**f**) and [Substrate]/K$_M$ (**g**). Data represent means (*n* = 2 independent experiments). **h, i** Selectivity of KMI169 against the full panel of SET-domain (**h**) and Rossmann-fold (**i**) methyltransferases available at Reaction Biology. Data represent means (*n* = 2 independent experiments).

upon increasing SAM or substrate concentrations corroborating a competitive inhibition mode (Fig. 3f, g).

To validate target selectivity of KMI169, we tested the compound against the full panel of SET domain- and Rossmann fold-containing methyltransferases available at Reaction Biology. The results showed that the compound was selective with the potential exception of PRMT5 (Fig. 3h, i). To investigate potential effects of KMI169 on PRMT5 activity, we performed enzymatic inhibition assays (Supplementary Fig. 4a).

Even at the highest concentration of 30 μM, reaching the limit of KMI169 solubility, we only observed about 60% maximum inhibition. For this reason, we could only calculate an approximate $IC_{50}$ value of $> 2$ μM (Supplementary Fig. 4a). Since this value is well above the $IC_{50}$ of 0.05 μM determined for KMT9 inhibition (Fig. 3e), KMI169 fulfils the strict criteria of a 'chemical probe' as defined by Arrowsmith et al.[22] and the Structural Genomics Consortium (https://www.sgc-unc.org/main-st).

To further exclude that KMI169 affected PRMT5 in cells, we analysed the levels of symmetric dimethyl arginine (SDMA)[23] in PC-3M cells cultured in presence of KMI169, KMI169Ctrl, or the PRMT5 inhibitor JNJ-64619178. As shown in Supplementary Fig. 4b, neither KMI169 nor KMI169Ctrl affected the levels of SDMA, whereas JNJ-64619178 abolished PRMT5-mediated arginine methylation. Together, these data demonstrate that under conditions of full KMT9 inhibition, PRMT5 activity is not detectably affected by KMI169. In addition, a KINOMEscan™ screen against a panel of 97 representative kinases from different families confirmed selectivity of KMI169 for KMT9 (Supplementary Fig. 4c). Taken together, we developed a high-affinity, selective KMT9 inhibitor (KMI169) and the related control compound (KMI169Ctrl) allowing us to validate chemical inhibition of KMT9 as an anticancer strategy in cellular assays.

## Inhibition of KMT9 impairs tumour cell proliferation

To assess cellular target engagement of KMI169, Cellular Thermal Shift Assays (CETSA)[24] were performed in human castration-resistant prostate cancer cell lines PC-3M and LNCaP-abl, which revealed strong stabilization of endogenous KMT9 ($\Delta T_m = 9.8$ K and 8.3 K, respectively) compared to KMI169Ctrl ($\Delta T_m = 1.0$ K) or DMSO (Fig. 4a, b and Supplementary Fig. 5a, b). In contrast, under these conditions neither KMI169 nor KMI169Ctrl stabilized PRMT5 (Supplementary Fig. 5c, d) further corroborating our conclusion that KMI169 does not target cellular PRMT5. Together, these results demonstrated cell membrane permeability and cellular target engagement of KMI169 making the inhibitor a suitable tool to study cellular KMT9 functions.

To address cellular consequences of KMT9 inhibition, we first treated different prostate cancer cell lines with KMI169. Results were compared with our previous data obtained upon KMT9 knockdown[10]. Firstly, when cells were treated with KMI169, we observed reduced levels of H4K12me1 (Fig. 4c and Supplementary Fig. 5e), which is in accordance with previous knockdown data[10]. In comparison, KMI169 did not alter histone marks such as H3K4me2, H3K9me2, and H4K20me1 (Supplementary Fig. 5e). Similarly, treatment of cells with the negative control compound KMI169Ctrl had no effect on H4K12me1 levels (Fig. 4c). Next, compared to vehicle (DMSO), KMI169 treatment strongly suppressed proliferation of PC-3M cells with a half-maximal growth inhibition ($GI_{50}$) of 150 nM (Fig. 4d). In contrast, under these conditions, KMI169Ctrl did not affect PC-3M cell proliferation (Fig. 4d). The anti-proliferative effect mediated by KMI169 was not restricted to PC-3M cells but also observed in other prostate, bladder, lung, and colon tumour cells, which is in accordance with previous knockdown data[10–12] (Fig. 4e).

In addition, we analyzed the effect of KMI169 in human hepatocellular carcinoma (HepG2) and pancreatic carcinoma (PANC-1) cells. Proliferation of both cell lines was previously reported to be insensitive to KMT9 depletion[10]. Despite cellular target engagement in both cell lines (Supplementary Fig. 5f-i), KMI169 did not affect proliferation of HepG2 or PANC-1 cells at concentrations that were effective in responsive cell lines (Fig. 4e and Supplementary Fig. 6a, b). Together, our data define KMI169 as a selective KMT9 inhibitor with cellular activity. Importantly, KMI169 treatment also impairs proliferation of castration- and enzalutamide-resistant prostate cancer cell lines such as PC-3M, DU145, LNCaP-abl, and LNCaP-abl EnzaR (Fig. 4e) hinting at the possibility to develop novel strategies for the treatment of prostate cancer.

## KMT9 inhibition affects expression of genes involved in cell cycle regulation

We previously showed that KMT9 controls the transcription of target genes involved in cell cycle regulation, thereby controlling prostate cancer growth[10]. To investigate whether KMT9 inhibition affected gene expression, we performed a global transcriptome (RNA-seq) analysis of PC-3M cells treated with KMI169 or DMSO control. We identified 1467 differentially expressed genes (DEGs), of which 801 were up- and 666 downregulated (Fig. 4f). In contrast, when PC-3M were cultured in the presence of KMI169Ctrl, only 20 DEGs were observed by RNA-seq (Supplementary Fig. 6c). Comparison of differential gene expression observed upon KMT9 inhibition (1467 DEGs) and previously reported KMT9 knockdown (6326 DEGs)[10] revealed an overlap of 897 DEGs in both datasets (Fig. 4g). These data suggest that several pathways may be commonly deregulated upon KMT9 inhibition and KMT9 knockdown. Deregulation of pathways that are not common under both conditions might be explained, for example, by potential scaffolding functions of KMT9 that are not affected by inhibitors of KMT9 catalytic function.

In PC-3M cells, KMT9 occupied 4729 target genes as previously shown by ChIP-seq[10] (Fig. 4h). Intersection with the 1467 DEGs observed upon KMT9 inhibition identified 382 DEGs with KMT9 gene occupancy (Fig. 4h). Gene enrichment analyses for these 382 direct target genes uncovered gene sets involved in cell cycle regulation (Fig. 4i, j). Quantitative RT-PCR analyses of PC-3M cells corroborated reduced mRNA levels upon KMI169 treatment for a selection of these genes including MYB proto-oncogene (*MYB*), aurora kinase B (*AURKB*), forkhead box A2 (*FOXA2*), cyclin dependent kinase 2 (*CDK2*), baculoviral iap repeat containing 5 (*BIRC5*), E2F transcription factor 1 (*E2F1*), E2F transcription factor 8 (*E2F8*), cell division cycle 6 (*CDC6*), and DNA ligase 1 (*LIG1*) (Fig. 4k). In contrast, mRNA levels of these genes were unchanged in PC-3M cells treated with KMI169Ctrl (Supplementary Fig. 6d). In addition, mRNA levels of these genes were unchanged in control HepG2 cells treated with KMI169 or KMI169Ctrl (Supplementary Fig. 6e). Collectively, our data show that KMT9 inhibition and knockdown target similar pathways resulting in deregulated expression of genes involved in cell cycle control, thereby accounting for impaired tumour cell proliferation.

Taken together, our data validate KMT9 as a potentially druggable target. KMI169 is a selective small molecule KMT9 inhibitor with cellular activity allowing to unravel biological functions of KMT9. Importantly, KMI169 is a promising starting point for the future development of clinical candidate KMT9 inhibitors, which might enable therapeutic strategies for the treatment of prostate cancer.

## Discussion

In this study, we developed KMI169, a potent and selective KMT9 inhibitor with cellular activity. KMI169 strongly impairs proliferation of prostate cancer cell lines by affecting the expression of KMT9 target genes involved in cell cycle control. To design KMI169, we followed a bi-substrate inhibitor strategy, which was previously successfully applied to develop other PMT inhibitors[25–36] including TC-5115 for MLL1[34], prodrug inhibitor SKI-73 for CARM1[32], and a 5,5-bicyclic nucleoside-derived PRMT5 inhibitor[36]. Our goal to design a potent and

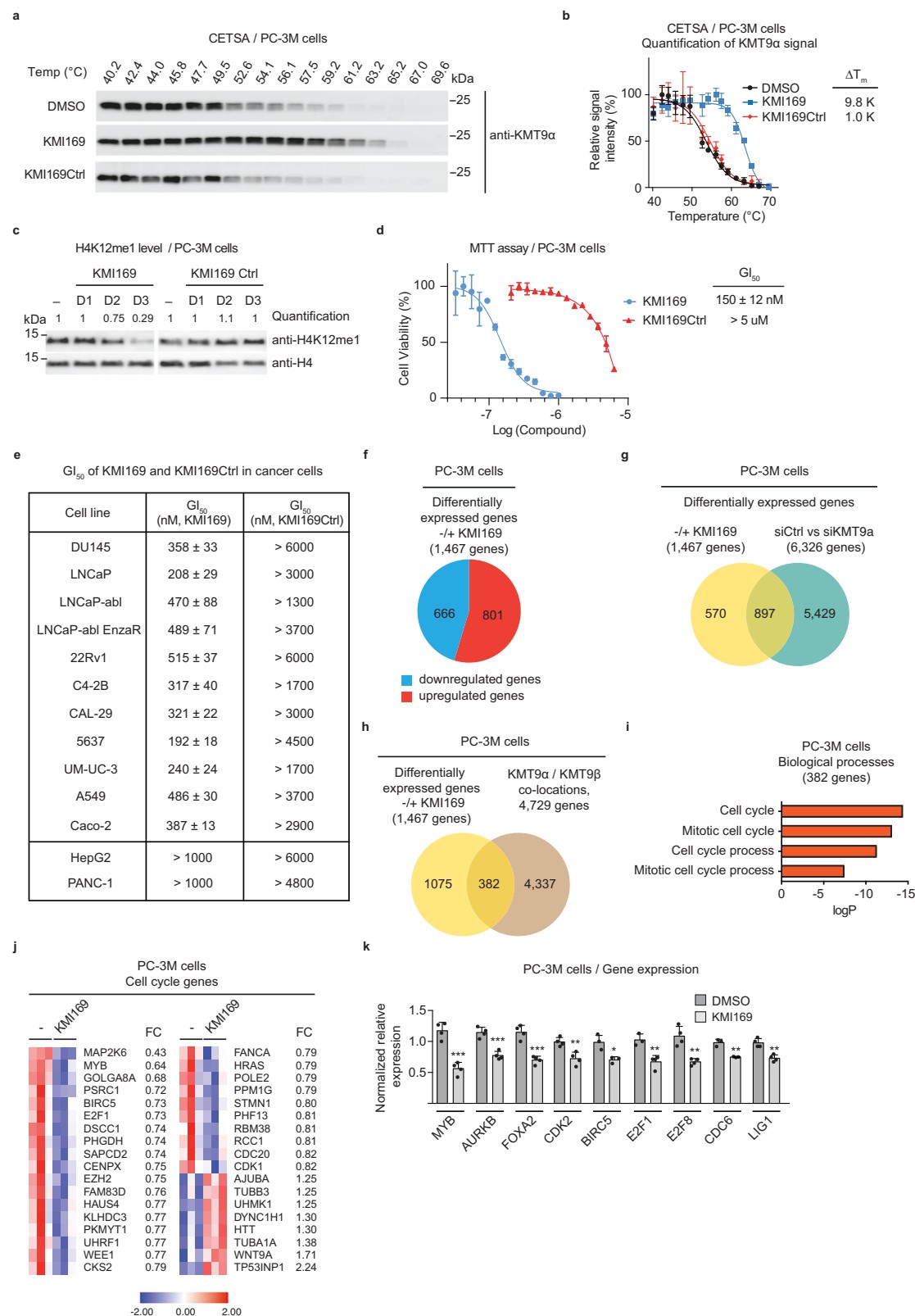

selective KMT9 inhibitor was reached by extensive modification of our initial hit (compound **1a**). Key features contributing to high affinity KMT9 binding include the azetidine amine anchor binding into the substrate channel and the 2-chloro extension of the adenosine mimic occupying sub-pocket A of KMT9.

Previously, a N-terminal methyltransferase inhibitor (NAH-C3-GPKK) was reported also to bind to KMT9 with moderate affinity[37].

NAH-C3-GPKK is a bi-substrate peptide inhibitor mainly consisting of a SAM and a GPKK peptide moiety and has only been tested in vitro. Cell membrane permeability and intracellular target engagement have not been evaluated, however, due to its unfavourable size and charge, the molecule is unlikely to enter cells[31,38]. In comparison, KMI169 fulfils all criteria of a 'chemical probe' as defined by Arrowsmith et al.[22] and the Structural Genomics Consortium (https://www.sgc-unc.org/main-st).

**Fig. 4 | Inhibition of KMT9 impairs tumour cell proliferation. a, b** CETSA for KMT9 in PC-3M cells treated with vehicle (DMSO), 1 μM KMI169 or KMI169Ctrl. Representative Western blots (**a**) and quantification (**b**) showing increased melting temperatures ($\Delta T_m$) of endogenous KMT9 upon treatment with KMI169 compared to KMI169Ctrl or DMSO. Data represent means ±s.d ($n = 3$ biologically independent experiments). **c** Levels of H4K12me1 in PC-3M prostate tumour cells cultured in the presence of DMSO (-), 500 nM KMI169 or 500 nM KMI169Ctrl for one (D1), two (D2) or three (D3) days were analysed by Western blot using the indicated antibodies. Histone H4 was used as control. **d** Concentration of half-maximal growth inhibition ($GI_{50}$) determined by MTT cell viability assays for KMI169 and KMI169Ctrl in PC-3M cells. Data represent means ± s.d ($n = 3$ biologically independent samples). **e** Table summarizing $GI_{50}$ values for KMI169- and KMI169Ctrl-mediated inhibition of proliferation of different tumour cells determined by MTT assays. Data represent means ± s.d ($n = 3$ biologically independent samples). **f** Diagram showing the number of differentially expressed genes (DEGs) in PC-3M cells treated four days with 360 nM KMI169 (+) or DMSO (-). **g** Venn diagram showing the intersection of DEGs upon treatment with KMI169 or RNAi-mediated knockdown of KMT9α. A one-sided hypergeometric test was used to calculate the significance of the overlaps ($p = 5.6 \times 10^{-144}$, $r = 2.0$, r: representation factor) **h** Venn diagram showing the intersection of DEGs upon treatment of PC-3M cells with KMI169 and KMT9α/β gene occupancy determined by ChIP-seq. A one-sided hypergeometric test was used to calculate the significance of the overlaps ($p = 4.924 \times 10^{-5}$, $r = 1.2$). **i** Enriched pathway analysis for 382 genes differentially expressed upon KMI169 treatment and showing KMT9 gene occupancy. Bars represent individual Benjamini $p$-values derived from GO enrichment analysis. **j** Heatmap displaying mRNA levels of differentially expressed genes in PC-3M cells cultured in the presence of DMSO (-) or 360 nM KMI169. FC: fold change. **k** QRT-PCR analysis showing relative mRNA levels of the indicated genes in PC-3M cells cultured in the presence of DMSO or 360 nM KMI169. Data represent means +s.d. *MYB* ($n = 4$, $p = 0.0003$), *AURKB* ($n = 4$, $p = 0.0005$), *FOXA2* ($n = 4$, $p = 0.004$), *CDK2* ($n = 4$, $p = 0.0053$), *BIRC5* ($n = 3$, $p = 0.0161$), *E2F1* ($n = 3$, $p = 0.0058$), *E2F8* ($n = 4$, $p = 0.0024$), *CDC6* ($n = 3$, $p = 0.002$), *LIG1* ($n = 4$, $p = 0.0014$). ($n$ represents the number of biologically independent samples. *$p < 0.05$, **$p < 0.005$, ***$p < 0.001$ by two-tailed Student's test.

The comparison of cellular pathways affected by KMT9 inhibition (this study) with previous data observed upon KMT9 depletion[10–12] revealed a strong overlap of genes that were differentially regulated under both conditions. Thus, the catalytic function of KMT9 appears to control similar cellular processes involved in prostate cancer cell proliferation. Since there is no full overlap between DEGs upon inhibition or depletion, KMT9 appears to possess additional, e.g. scaffolding functions contributing to cellular KMT9 activity. Partial overlaps for DEGs upon inhibitor treatment and knockdown of the gene of interest appear to be commonly observed and have been documented in previous publications[39–41]. In addition to catalytic vs. scaffolding functions of KMT9, differing experimental conditions optimised for the respective purpose (in our study 96 h of inhibitor treatment vs. 72 h of siRNA treatment) or distinct efficiencies of inhibition and knockdown might contribute to the partial overlap. Finally, it cannot be excluded that gene expression changes upon KMT169 treatment are not solely caused by KMT9 inhibition.

PMTs are involved in various human diseases including cancer and thus promising therapeutic targets. Since the discovery of the first inhibitor (chaetocin) of a histone lysine methyltransferase[42], tremendous progress has been made in the development of PMT inhibitors. For instance, the enhancer of zeste 2 (EZH2) inhibitor EPZ6438 (Tazemetostat, TAZVERIK™) has been approved for the treatment of follicular lymphoma and epithelioid sarcoma, and EPZ5676 (Pinometostat), an inhibitor targeting DOT1L, as well as PRMT5 inhibitors (e.g. JNJ-6461978 and GSK-3326595) have entered clinical trials[1].

The anti-proliferative effect of KMI169 documented in this study provides evidence that inhibition of KMT9 catalytic activity is capable to impair cancer cell growth. Notably, our KMT9 inhibitor halted proliferation of castration- and enzalutamide- resistant prostate cancer cells. Therefore, our results not only provide a proof-of-concept for the druggability of KMT9 but also introduce KMT9 inhibitors as potential candidates for novel treatment options for currently incurable disease states such as castration- and enzalutamide-resistant prostate cancer.

## Methods
### Synthesis of chemical compounds
Synthesis and spectroscopic characterization of compounds are described in Supplementary Methods.

### Plasmids
For plasmid construction, the cDNA fragments corresponding KMT9α, KMT9β and ETF1 obtained from GeneCopoeia were cloned into pET-Duet1 and pDEST17 as previously described[10].

### Protein expression and purification
Plasmid construction, expression, and purification of the recombinant KMT9 heterodimer and His-ETF1aa141-275 were performed as described previously[10]. Briefly, for protein expression of KMT9 heterodimer and ETF1, Pet-Duet1-KMT9α–His-KMT9β and pDEST17-ETF1-141-275 were transformed into BL21-CodenPlus-RIPL. Protein expression was performed overnight at 18 °C in the presence of 0.4-1 Mm IPTG. Bacterial pellets were resuspended in binding buffer 1 and disrupted with EmulsiFlex (Avestin) and lysates were clarified via centrifugation. The protein was purified by Ni-NTA column (Qiagen) followed by anion exchange purification and gelfiltration. For crystallization, Pet-Duet1-His-TEV-KMT9α-16-218-KMT9β was expressed and purified in a same way except that the His tag was cleaved off by TEV protease digestion overnight during dialysis. Protein was concentrated to 25 mg/ml in gel filtration buffer, aliquoted, flash frozen in liquid nitrogen, and stored at -80 °C.

### Microscale thermophoresis (MST)
To determine KMT9/compound binding affinities, MST was performed using a Monolith NT.115 instrument (NanoTemper Technologies GmbH). His-tagged KMT9 was labelled using the His-Tag Labeling Kit RED-tris-NTA 2nd Generation (NanoTemper Technologies GmbH) according to the manufacturer's instructions. Labelling and binding assays were performed in buffer containing 25 mM HEPES (pH 7.5), 100 mM NaCl, 1 mM DTT, and 0.05% Tween. Compound 1:1 dilution series were prepared, mixed with labelled KMT9 protein (20 nM final concentration), and incubated for 30 min at room temperature. After centrifugation, samples were loaded into standard capillaries (NanoTemper Technologies GmbH), and MST measurements were performed at 40% MST and 100% LED power using the MO.Control program. Datasets were processed with the MO. Affinity analysis software (NanoTemper Technologies GmbH).

### KMT9 methyltransferase inhibition assays
The assay was carried out in duplicates in assay buffer (50 mM BTP (pH 8.5), 1 mM MgCl2, 1 mM DTT and 0.01% Triton-X100) in the presence of inhibitor, 25 nM KMT9, 0.3 μM $^3$H-SAM, 0.7 μM SAM, and 5 μM His-ETF1aa140-275 in a final volume of 20 μl. Reactions were incubated at 30 °C for 2 h with shaking and then stopped by adding 5 μl of a 50% trichloroacetic acid (TCA) solution. The 22 μl reaction mixtures were transferred into 96-well MultiScreen®HTS FB filter plates (Merck) and subsequently washed with 10% TCA and 100% ethanol. After drying overnight, filters were transferred into Pony Vials (Perkin Elmer Inc.) and incubated in 3 ml of Ultima Gold scintillation cocktail (Perkin Elmer Inc.) for 30 min. The scintillation signal was measured 3 times for

1 min using a TriCarb 2910 TR (Perkin Elmer Inc.) scintillation counter set to $^3$H CPM mode (LL: 0, UL: 18.6).

Inhibition of KMT9 activity was calculated using the following formula:

$$\text{Inhibition [\%]} = \left(1 - \frac{x_c - x_{pos}}{x_{pos} - x_{neg}}\right) * 100$$

With $x_c$: signal of compound, $x_{pos}$: mean signal of positive control, $x_{neg}$: mean signal of negative control.

## Cofactor/substrate competition assays

Cofactor competition experiments were performed in presence of 25 nM KMT9, 5 μM ETF1 in assay buffer (50 mM bis-tris propane, 1 mM MgCl2, 1 mM DTT and 0,01% Triton-X100 at a pH of 8.5). Five different SAM concentrations were adapted (0.25 μM, 1 μM, 2 μM, 4 μM and 8 μM, 30% 3H-SAM for each concentration) to obtain five IC$_{50}$ values. Substrate competition experiments were performed in presence of 25 nM KMT9, 0.3 μM $^3$H-SAM, 0.7 μM SAM and 0.625, 2.5, 5, 10, or 20 μM ETF1 to obtain five IC$_{50}$ values. The assay was conducted and analysed as described in "KMT9 methyltransferase inhibition assays".

## Fluorescence Thermal Shift Assay (FTSA)

The assay was carried out in 96-well hard-shell PCR plates (Bio-Rad Laboratories Inc.). Assay buffer contained 50 mM BTP (pH 8.5), 1 mM MgCl$_2$, and 1 mM DTT. 1 μl of inhibitor (20x stock in DMSO) or 1 μl of DMSO (vehicle control), 10 μl of KMT9 (2x stock in assay buffer), 5 μl of SyproOrange (20x stock in assay buffer), and 4 μl of assay buffer were mixed (total volume: 20 μl). Final concentrations in the assay were: 500 μM (compound **1a** / **1b** / **1c**) or 10 μM inhibitor (compounds **2a-h**, **3a**, **3b**, KMI169, and KMI169Ctrl), 2 μM KMT9, and 5x SyproOrange. The PCR plate was centrifuged at 88 × g for 1 min in a UNIVERSAL 320 centrifuge (Hettich) and incubated at 25 °C for 15 min with shaking. After incubation, the plate was again centrifuged for 1 min at 88 × g, and measurement was conducted using a CFX96 Touch Real-Time PCR Detection System (Bio-Rad Laboratories Inc.). The plate was equilibrated at 20 °C for 4 min and then heated stepwise at a rate of 1 °C per 15 s until 95 °C were reached. After every step, fluorescence was measured ($\lambda_{excitation}$ = 485 nm, $\lambda_{emission}$ = 530 nm) in 'FRET mode'. Calculation of melting points was conducted using a Boltzmann sigmoidal model in GraphPad Prism 7.0.

## Crystallisation, data collection, and structure determination

For crystallisation, purified KMT9 protein was mixed with a 10-fold excess of compound **1a** / **1b** / **1c**. Crystals grew in 1.3 M Na$_3$Citrate, 0.1 M Tris (pH 8.0) at 20 °C, appeared after one day, and reached full size within one week. Crystals were cryoprotected with 10% glycerol and flash frozen in liquid nitrogen. Data were collected at the Swiss Light Source beamline PX3 using a wavelength of 1.0000 Å at 100 K. Data were processed and analysed with XDS-20210323[43] and Aimless-0.7.2[44]. Crystal belonged to the space group P6$_1$22 containing one KMT9 heterodimer molecule per asymmetric unit. The structure was solved by molecular replacement with Phaser[45] using the published KMT9 structure (PDB code 6H1D) as a search model. Manual building and refinement were performed using Coot-0.9.8.4[46] and Refmac-5.8.0352[47] in the CCP4-8.0.0005 package[48]. The final models were validated using MolProbity-4.5.2[49] in the Phenix-1.20rc1-4387 package[50] and the RCSB Validation server. Supplementary Table 1 summarises the data collection and refinement statistics. Crystallographic data have been deposited at the Protein Data Bank under the PDB code 8CNC, 8QDG, and 8QDI.

## Structure based docking and modelling

Schrodinger suite 2019-1 (Schrodinger, NY) was used for modelling. The KMT9/compound **1a** co-crystal structure (PDB code 8CNC) was used as receptor and prepared with the Protein Preparation Wizard. Ligands (compounds **3b** and KMI169) were prepared and minimized using the Ligprep Module. Glide SP[51] was used for the docking with default settings. Prime[52] MM-GBSA was performed by using the docking poses from Glide SP.

## Methyltransferase selectivity assays

The methyltransferase selectivity profile of KMI169 was evaluated by testing the inhibition level for all SET domain- and Rossmann fold-containing methyltransferases available at Reaction Biology. Inhibition was assessed with KMI169 concentrations of 1.5 μM in a radiometric assay measuring substrate methylation using $^3$H-labelled SAM. The selectivity assay was performed by Reaction Biology, Malvern, USA.

## Kinase selectivity assay

The kinase selectivity profile of KMI169 at 10 μM was validated by the KINOMEscan$^{TM}$[53] Profiling Service performed at Eurofins DiscoverX Corporation, San Diego, USA. Compound-kinase interactions were tested with 97 representative kinases belonging to the AGC, CAMK, CMGC, CK1, STE, TK, TKL, lipid, and atypical kinase families including important mutant forms (scanEDGE$^{TM}$ Kinase Panel).

## Cell lines

HepG2 (#HB-8065), 5637 (#HTB-9), UM-UC-3 (#CRL-1749), A549 (CRM-CCL-185), Caco-2 (HTB-37), PANC-1 (CRL-1469), DU145 (HTB-81), C4-2B (#CRL-3315), and 22Rv1 (#CRL-2505) cells were obtained from ATCC. CAL-29 (#ACC 515) cells were obtained from DSMZ. PC-3M (CVCL_5J25) and LNCaP (CVCL_5J24) cells were obtained from Caliper Life Science. LNCaP-abl (CVCL_4793) and LNCaP-abl-EnzaR cells were obtained from Z. Culig. Cell lines were tested for mycoplasma and found to be uncontaminated. None of the used cell line is known as misidentified cell line by the International Cell Line Authentication Committee.

## Cell culture

PC-3M, LNCaP, LNCaP-abl, LNCaP-abl-EnzaR, C4-2B, and 5637 cells were cultured in RPMI 1640. CAL-29, HepG2, 22Rv1, PANC-1, Caco-2, and A549 cells were cultured in DMEM. UM-UC-3 and DU145 cells were cultured in EMEM. All media were supplemented with 10% foetal calf serum, penicillin/streptomycin, and glutamine. The culture medium for HepG2 cells was supplemented with non-essential amino acids, medium for LNCaP-abl-EnzaR cells was supplemented with 13 μM enzalutamide.

## Western blot analysis

Histones used for Western blot analysis were extracted from cells as follows. Cells were cultured in the presence of vehicle (DMSO), KMI169 (500 nM; Fig. 4c and Supplementary Fig. 5e), or KMI169Ctrl (500 nM; Fig. 4c) for three days. Cells were harvested, washed with PBS, re-suspended in Triton Extraction Buffer (TEB; PBS, Triton-X100, 2 mM PMSF, 0.02% NaN$_3$), incubated 10 min on ice, centrifuged, and the supernatant was discarded. After washing cells, a second time with TEB, pellets were resuspended in 0.2 M HCl, and histones were acid-extracted over night at 4 °C. Upon centrifugation for 10 min at 425 x g, supernatant was collected and protein concentration was determined. For Western blot, the following antibodies were used: anti-KMT9α (#27630, lot 20062017, Schüle Lab; 1/1000), anti-histone H4 (#ab10158, lot GR322677-1, Abcam, 1/3000), anti-H4K12me1 (#27429, lot 27062017, Schüle Lab, 1/1000), anti-PRMT5 (#MBS9405987, lot 6018, MyBioSource, 1/1000), anti-histone H3 (#ab1791, lot GR300976-1, Abcam, 5000), anti-H3K4me2 (#CS-35-100, lot A391-001, Diagenode, 1/750), anti-H3K9me2 (#07-441, lot 1463717, Millipore, 1/1000), anti-H4K20me1 (#39727, lot 21115004, Active Motif, 1/1000), anti-symmetric dimethyl arginine motif (SDMA, #13222, lot 8, Cell Signalling, 1/1000), anti-GAPDH

(#MAB374, lot 3688975, Millipore, 1/500). Validation data for anti-KMT9α and anti-H4K12me1 antibodies are described in Ref. 10. Validation information for anti-GAPDH, anti-SDMA, anti-histone H4, anti-PRMT5, anti-histone H3, anti-H3K4me2, anti-H3K9me2, anti-H4K20me1 antibodies are described in the reporting summary. Chemoluminescent signals were recorded with an Amersham imager 600 (GE Healthcare) and quantified using Amersham imager 600 1.2.0 software.

## MTT assay

Cell proliferation was determined using the CellTiter 96® Non-Radioactive Cell Proliferation Assay (MTT) kit (Promega) essentially as described by the manufacturer. PC-3M (300 cells/well), LNCaP (2500 cells/well), LNCaP-abl (5000 cells/well), LNCaP-abl-EnzaR (2500 cells/well), 22Rv1 (5000 cells/well), C4-2B (750 cells/well), DU145 (375 cells/well), CAL-29 (1000 cells/well), 5637 (550 cells/well), UM-UC-3 (700 cells/well), A549 (480 cells/well), Caco-2 (800 cells/well), PANC-1 (3000 cells/well), or HepG2 (6000 cells/well) were seeded in 96 well plates in the presence of DMSO, KMI169 or KMI169Ctrl at different concentrations and allowed to grow for seven days prior to MTT measurement. Cell culture medium containing DMSO, or inhibitor was refreshed at day 4. Calculation of the $GI_{50}$ was conducted using a sigmoidal model in GraphPad Prism 6.0.

## RNA preparation for RNA-seq and analysis

PC-3M cells were cultured in the presence of DMSO or 360 nM KMI169 or 360 nM KMI169Ctrl for four days. RNA was isolated using the RNeasy Plus Mini kit (Qiagen) essentially as described by the manufacturer. RNA samples were sequenced using the standard Illumina protocol generating raw sequence files (.fastq files) by Novogene. Reads were aligned to the hg19 build of the human genome using STAR version 2.7.10b. The aligned reads were counted with the Homer 4.11 software with default settings (analyzeRepeats) and differentially expressed genes were identified using EdgeR 2.38.4. Heatmaps were generated using Morpheus (https://software.broadinstitute.org/morpheus/). Data are deposited under GSE235595 [https://www.ncbi.nlm.nih.gov/geo/query/acc.cgi?acc=GSE235595].

## RNA preparation for quantitative RT-PCR and analysis

Cells were cultured in the presence of DMSO, 360 nM KMI169 or KMI169Ctrl for four days. RNA was isolated using the RNeasy Plus Mini kit (Qiagen) essentially as described by the manufacturer. (in detail) Quantitative RT-PCR was performed using the Abgene SYBR Green PCR kit (Invitrogen) according to the supplier's protocol. *GAPDH* was used for normalization. The following primers were used: *GAPDH*: 5′-GAGTCCACTGGCGTCTTCAC-3′ and 5′-GTTCACACCCATGACGAACA-3′; *MYB*: 5′-GGGCAGAAATCGCAAAGCTA-3′ and 5′-GGCAGGGAGTTGAGCTGTA-3′; *AURKB*: 5′-TGTGTGGCACCCTGGACTAC-3′ and 5′-GGAAGCGGGGAACTTTAGGT-3′; *FOXA2*: 5′-TGCACTCGGCTTCCAGTATG-3′ and 5′-CGTGTTCATGCCGTTCATCC-3′; *CDK2*: 5′-TTGTCAAGCTGCTGGATGTC-3′ and 5′-TGATGAGGGGAAGAGGAATG-3′; BIRC5: 5′-GGACCACCGCATCTCTACAT-3′ and 5′-GAAACACTGGGCCAAGTCTG-3′; *E2F1*: 5′-CATCATCTCCCCCCTCATC-3′ and 5′- GAGGCCGGAGAAGTCCTC-3′; E2F8: 5′-GCCTAGAAGTTGCTGCCAAG-3′ and 5′-TTTCGGCCTCTTTCCTCTGT-3′; CDC6: 5′-CCAAAAAGGAAGCTGTCTCG-3′ and 5′-CAGGGCTTTTACACGAGGAG-3′; LIG1: 5′-ACAAATATGACGGGCAGAGG-3′ and 5′-GCTGATGATGTCCGGGTACT-3′. Customs primers were obtained from Merck.

## Cellular Thermal Shift Assay (CETSA)

For CETSA, PC-3M, LNCaP-abl, PANC-1, and HepG2 cells were incubated with 1 μM of KMI169 or 1 μM KMI169Ctrl for 24 h. Then, cells were washed, harvested using trypsin, and pelleted by centrifugation. The pellet was washed twice with PBS supplemented with cOmplete™ EDTA-free Protease Inhibitor Cocktail (Roche) and Phosphatase

Inhibitor Cocktail 2 and 3 (Sigma) and divided into 16 aliquots of 40 μl containing 150,000 cells (PC-3M), 145,000 cells (LNCaP-abl), 160,000 cells (PANC-1), or 200,000 cells (HepG2). Cells were then frozen in liquid nitrogen and thawed at 25 °C three times and each aliquot was heated to a defined temperature by applying a gradient between 40 °C and 70 °C in a PCR cycler (Mastercycler Nexus Gradient, Eppendorf; 16-well gradient, 55 °C ± 15). After 3 min incubation, aliquots were snap frozen in liquid nitrogen and thawed at 25 °C. Cell lysates were centrifuged at 20000x g (4 °C) for 20 min. Supernatants were mixed with 5x loading buffer, heated to 95 °C for 5 min, and analysed by Western blotting. Chemoluminescent signals were recorded with an Amersham imager 600 (GE Healthcare) and quantified using Amersham imager 600 1.2.0 software. Calculation of ΔTm was conducted using sigmoidal model in GraphPad Prism 6.0.

## Statistics and reproducibility

In general, at least two independent experiments were performed with similar results. No statistical method was used to predetermine sample size. No data were excluded from the analyses. The experiments were not randomized. The Investigators were not blinded to allocation during experiments and outcome assessment. Data are represented as mean + or +/- standard deviation (s.d) or standard error of the mean (s.e.m) as indicated. Significance was calculated by two-tailed Student's *t*-tests or one-sided hypergeometric test as indicated in the figure legends. Statistical significance was set to $P < 0.05$ and is represented as follows: ***$p < 0.001$, **$p < 0.01$, *$p < 0.05$. Sample sizes are indicated where appropriate.

## Reporting summary

Further information on research design is available in the Nature Portfolio Reporting Summary linked to this article.

## Data availability

The authors declare that the data supporting the findings of this study are available within the article and its Supplementary Information files, or are available on request. Crystallographic data have been deposited at the Protein Data Bank under the PDB code 8CNC, 8QDG, and 8QDI. The RNAseq data are deposited under GSE235595. The accession codes of all other PDB coordinate files referenced in this study are: 6H1E and 2RFI. Source data are provided with this paper.

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

## Acknowledgements

We thank S. Günther and P. Mishra (Institute of Pharmaceutical Sciences, University of Freiburg) for their support in designing the hit compound. We are obliged to B. Breit and P. Regenass (Institute of Organic Chemistry, University of Freiburg) for assisting the initial synthesis of the hit compound and A.A. Baniahmad and V.I. Hazai (Institute of Pharmaceutical Sciences) for supporting chemical syntheses. We are grateful to Swiss Light Source (SLS) beamline scientists and N. Schmidt for

technical support. This work was supported by grants of the Deutsche Forschungsgemeinschaft (DFG, German Research Foundation): SFB 1381 (Project-ID: 403222702), Schu688/15-1, and Schu688/18-1 to R.S., SFB 992 (Project-ID: 192904750), to R.S., O.E. and M.J., and CIBBS Germany´s Excellence Strategy—EXC-2189—Project ID: 390939984 to M.J. and R.S. E.M. was supported by grant DKTK FR01-374. K.M.S was funded by the EQUIP Program for Medical Scientist, Faculty of Medicine, University Freiburg and a SFB 992 (Project-ID: 192904750) fellowship. We are deeply grateful for the charitable contributions of G. Rockstroh.

## Author contributions

R.S., E.M., M.J., C.M., C.G., C.G.T. and S.W. generated the original hypothesis. S.W., E.M., S.U., J.B., N.P.F.B., L.P., S.O.K., K.M.S., G.A.R., L.Z., O.E., H.B., M.SU. and M.S. performed experiments. D.W. performed bioinformatics analyses. S.W., H.G., E.M., M.J. and R.S. took primary responsibility for writing the manuscript. All authors edited the manuscript.

## Funding

## Competing interests

The authors declare no competing interests.
