## [Peer Review File · Nature Communications]

Structure-guided design of a selective inhibitor of the methyltransferase KMT9 with cellular activityEditorial Note: This manuscript has been previously reviewed at another journal that is not operating a transparent peer review scheme. This document only contains reviewer comments and rebuttal letters for versions considered at Nature Communications. Mentions of prior reports have been redacted.

Reviewer #1 (Remarks to the Author):

General:

The authors have addressed most of the raised points. Please consider to remove "first" in the title as it is slightly controversial.

1.) The authors claim 14 additional ligand-bound KMT9 structures of close analogues supporting their modeling hypothesis. At least one example should be deposited, discussed, and included in the supplementary section, it would be very helpful to follow the argumentation.

2.) and 3)

Most of the issues have been addressed. However, the Ctrl compound should be profiled in the proliferation assays side by side (Figure 4e). The two insensitive cell lines (HepG2 cells, PANC-1 cells) should be included in figure 4e.

Reviewer #2 (Remarks to the Author):

Wang et al. report in the manuscript "Structure-guided design of the first selective inhibitor of the methyltransferase KMT9 with cellular activity" the discovery of a novel inhibitor of the methyltransferase KMT9. and based on limited cancer cell viability and gene expression data, advertise it as the first cellular probe to dissect catalytic and non-catalytic function(s) of KMT9. While the described experiments are well executed and have merit, the advancement is incremental and the cell-based data too preliminary to warrant publication in Nature Communications.

1. The conceptual novelty is limited given that bi-substrate methyltransferase inhibitors for KMT9 have been reported previously.

2. In the revised manuscript the authors have strengthened data addressing methyltransferase selectivity. While PRMT5 seems to be also inhibited to a certain extent the CETSA and SDMA assessment experiments are convincing that there is a preference for KMT9 versus PRMT5 inhibition.

3. The claim about mechanism-based KMT169 induced growth phenotypes is preliminary. The authors state that cell lines that are independent of KMT9 (HepG2 and PANC1) are inert to KMT169 treatment. This does not necessarily validate that the KMT-169 induced growth phenotype in PC-3M cell is on target. The authors needed to "rescue" the growth phenotype (e.g. mutant version of KMT9 that is active but incapable of compound binding) to unambiguously demonstrate on mechanism phenotypic activity.

4. The questions around the limited overlap of KMT169-induced and KMT9 siRNA-induced gene expression changes remain. The authors use somewhat circular logic since they say the differences may be due to inhibition versus loss of protein by use of KMT169 and KMT9 specific siRNAs respectively. However, they use the concordance of KMT9 siRNA and KMT169 mediated growth phenotypes in PC-3M and the absence thereof in PANC1 and HepG2 as evidence for mechanism-based phenotypic effects. The authors have not sufficiently demonstrated that the KMT169 mediated gene expression changes are solely due to inhibition of KMT9 methyltransferase activity.

5. [REDACTED] in Fig 2E the interaction of the azetidino moiety to residue Y125 is highlighted as a key interaction, however mutation of which does not affect compound binding.

Reviewer #3 (Remarks to the Author):

The authors adequately addressed many of my concerns. I am now supportive of approving this manuscript for publication.

Reviewer #1 (Remarks to the Author):

General:

The authors have addressed most of the raised points. Please consider to remove "first" in the title as it is slightly controversial.

As suggested, we removed 'first' in the title (page 1, line 1) and throughout the text (page 4, line 12, page 10, line 24 and page 12, lines 4 and 11).

1.) The authors claim 14 additional ligand-bound KMT9 structures of close analogues supporting their modeling hypothesis. At least one example should be deposited, discussed, and included in the supplementary section, it would be very helpful to follow the argumentation.

We followed the reviewer's suggestion and deposited two representative and informative ligand-bound KMT9 structures (PDB ID: 8QDG and 8QDI), included them in the supplementary section (new Extended Data Fig. 2a-f) and describe the additional structures in the results (page 5, lines 21-28 and page 6, lines 1-3). Together, the ligand-bound KMT9 structures show an overall high rigidity of the pocket in the presence of analogues with different substrate moieties, which fully corroborates the modelling approach applied for our compound series 2 to 4.

2.) and 3)

Most of the issues have been addressed. However, the Ctrl compound should be profiled in the proliferation assays side by side (Figure 4e). The two insensitive cell lines (HepG2 cells, PANC-1 cells) should be included in figure 4e.

We profiled KMI169ctrl in the proliferation assay side by side and calculated the GI_{50} of KMI169 in HepG2 and PANC-1 cells. We included the data in the revised Fig. 4e. In the revised manuscript (page 10, lines 20-23) we state: "Despite cellular target engagement in both cell lines (Extended Data Fig. 4f-i), KMI169 did not affect proliferation of HepG2 or PANC-1 cells at concentrations that were effective in responsive cell lines (Fig. 4e, and Extended Data Fig. 4j, k)".

Reviewer #2 (Remarks to the Author):

Wang et al. report in the manuscript "Structure-guided design of the first selective inhibitor of the methyltransferase KMT9 with cellular activity" the discovery of a novel inhibitor of the methyltransferase KMT9. and based on limited cancer cell viability and gene expression data, advertise it as the first cellular probe to dissect catalytic and non-catalytic function(s) of KMT9. While the described experiments are well executed and have merit, the advancement is incremental and the cell-based data too preliminary to warrant publication in Nature Communications.

1. The conceptual novelty is limited given that bi-substrate methyltransferase inhibitors for KMT9 have been reported previously.

We acknowledge that previously a NTMT1 bi-substrate inhibitor was also found to bind KMT9. The major achievement of our study was to develop a potent and selective KMT9 inhibitor with cellular activity starting from an initial hit. In addition, we characterized this novel inhibitor in cellular assays allowing us to distinguish between catalytic and scaffolding functions of KMT9. In our opinion, these are novel and major advancements compared to the previously shown non-selective, non-cell permeable NTMT1 bi-substrate inhibitors, which are not suitable for the characterization of KMT9 biology.

2. In the revised manuscript the authors have strengthened data addressing

methyltransferase selectivity. While PRMT5 seems to be also inhibited to a certain extent the CETSA and SDMA assessment experiments are convincing that there is a preference for KMT9 versus PRMT5 inhibition.

We are happy to hear that our data convinced reviewer #2.

3. The claim about mechanism-based KMT169 induced growth phenotypes is preliminary. The authors state that cell lines that are independent of KMT9 (HepG2 and PANC1) are inert to KMT169 treatment. This does not necessarily validate that the KMT-169 induced growth phenotype in PC-3M cell is on target. The authors needed to “rescue” the growth phenotype (e.g. mutant version of KMT9 that is active but incapable of compound binding) to unambiguously demonstrate on mechanism phenotypic activity. The authors have not sufficiently demonstrated that the KMT169 mediated gene expression changes are solely due to inhibition of KMT9 methyltransferase activity.

We agree with reviewer #2. Our statement that HepG2 and PANC-1 are inert to KMI169 treatment does not necessarily validate that the KMI169-induced growth phenotype in PC-3M is on target.” For more clarity, we modified the text accordingly: “In addition, we analyzed the effect of KMI169 in human hepatocellular carcinoma (HepG2) and pancreatic carcinoma (PANC-1) cells.” (please see page 10, lines 18-19).

While a fully active KMT9 mutant that is incapable of compound binding would be a nice tool, we think that identification of such a mutant is a goal that most likely either cannot be achieved at all or cannot be achieved within a reasonable amount of time. The simple reason for our scepticism stems from the fact that our bi-substrate inhibitors, by design, utilize the same KMT9 contacts as cofactor (SAM) and substrate. Consequently, mutations that abolish binding of bi-substrate inhibitor will also interfere with the binding of SAM or substrate and compromise KMT9 activity. Therefore, while desirable, it would most likely be extremely difficult, if not impossible, to identify a KMT9 mutant with the desired properties.

Since such mutants are so difficult/impossible to generate, we (like other researchers involved in inhibitor development) applied a large panel of experimental approaches to show that KMI169 indeed acts on KMT9. To demonstrate that the KMI169-induced growth phenotype in PC-3M cells is specific and on target, we showed that KMI169 directly binds KMT9 in PC-3M cells (CETSA, Fig. 4a), we demonstrated a decrease in H4K12me1 levels at day 3 of treatment with KMI169 (Fig. 4c), and uncovered differential gene regulation in PC-3M cells (Fig. 4f). We further observed that cell cycle genes were downregulated upon treatment with KMI169 in PC-3M cells (Fig.4i-k).

In addition, we include below a comparative analysis showing that the vast majority cell cycle genes (116 out of 170; 68%) that are downregulated upon treatment with KMI169 are also downregulated upon RNAi-mediated KMT9 knockdown (please see Figure 1 below). These data further support the fact that the KMI169-induced growth phenotype in PC-3M cells is on target.

PC-3M cells

Downregulated cell cycle genes
-/+ KMI169 siCtrl vs siKMT9 α
(170 genes) (409 genes)

Figure 1: Venn diagram showing the intersection of cell cycle genes downregulated upon treatment of PC-3M cells with KMI169 and RNAi-mediated KMT9 α knockdown.

While individual experiments are certainly not sufficient to substantiate claims, the sum of all experiments and controls provides, in our opinion, strong evidence for our conclusions. Nevertheless, we acknowledged in the discussion “Finally, it cannot be excluded that gene expression changes upon KMT169 treatment are not solely caused by KMT9 inhibition” (page 13, lines 14-16).

4. The questions around the limited overlap of KMT169-induced and KMT9 siRNA-induced gene expression changes remain. The authors use somewhat circular logic since they say the differences may be due to inhibition versus loss of protein by use of KMT169 and KMT9 specific siRNAs respectively. However, they use the concordance of KMT9 siRNA and KMT169 mediated growth phenotypes in PC-3M and the absence thereof in PANC1 and HepG2 as evidence for mechanism-based phenotypic effects.

As [REDACTED] added in the discussion (page 13, lines 9-15), partial overlaps for DEGs after inhibitor treatment and knockdown of the gene of interest appear to be commonly observed and have been documented in previous publications. (for example: Extended Data Fig. 5a-b, Yankova, E. et al. Small-molecule inhibition of METTL3 as a strategy against myeloid leukaemia. *Nature* 593, 597-601 (2021); Fig. S11-F, J and Fig. S14-C, Yu, X. et al. A selective WDR5 degrader inhibits acute myeloid leukemia in patient-derived mouse models. *Sci. Transl. Med.* 13 (2021); Fig 5I, Wang, J. et al. EZH2 noncanonically binds cMyc and p300 through a cryptic transactivation domain to mediate gene activation and promote oncogenesis. *Nat. Cell Biol.* 24, 384-399 (2022).

The reasons for the limited overlap of KMI169- and KMT9 siRNA-mediated gene expression changes are multiple and might be due to differing experimental conditions optimized for the respective purpose (96 hours inhibitor treatment vs. 72 hours siRNA treatment). In fact, the response to inhibitor seems to be relatively slow since we only observed few differentially expressed genes after 72 hours of treatment. In addition, the efficiencies of knockdown and inhibition may differ. Assuming, for example, that KMT9 inhibition is very efficient, whereas the knockdown efficiency is 'only' in the range of e.g. 80-90%, the remaining KMT9 protein may cause gene expression not observed upon (efficient) inhibition (and vice versa).

Our entire chain of argumentation includes in vitro and in vivo selectivity tests, the use of a structurally related control compound (KMI169ctrl), assays in control cell lines, as well as comparisons of gene expression upon KMT9 inhibition and knockdown. While a single experiment would certainly not allow conclusions about KMI169 selectivity, the complete set of experiments substantiates our claims. We acknowledge, however, that we cannot exclude that a small percentage of differentially expressed genes might be the result of off-target activity of either siRNA or inhibitor. However, such potential 'side-effects' effects do not affect the overall conclusions of the manuscript. Accordingly, in the revised manuscript we now state “Finally, it cannot be excluded that gene expression changes upon KMT169 treatment are not solely caused by KMT9 inhibition” (page 13, lines 14-16).

5. [REDACTED] in Fig 2E the interaction of the azetidine moiety to residue Y125 is highlighted as a key interaction, however mutation of which does not affect compound binding.

Fig. 2E shows key interactions of KMT9 with KMI169 around the azetidine moiety. The figure depicts three main contacts in this region, while in other parts of the ligand binding pocket there are multiple additional protein-inhibitor contacts. Since Y125A compromises only one out of multiple ligand contacts, the two-fold reduced inhibitor binding of the mutant ($K_d = 50$ nM for KMI169 binding to Y125A vs. $K_d = 25$ nM for KMI169 binding to wildtype KMT9) lies actually within the expected range. We did not expect Y125A to drastically or fully abolish inhibitor binding as possibly assumed by the referee.

Reviewer #3 (Remarks to the Author):

The authors adequately addressed many of my concerns. I am now supportive of approving this manuscript for publication.

We thank reviewer #3 for his very positive judgement.

Reviewer #1 (Remarks to the Author):

The authors have addressed the raised points and I can recommend the manuscript for publication.

Reviewer #2 (Remarks to the Author):

The authors now included language in the revised manuscript that address my main concerns. I am now supportive of publication of this manuscript.

Reviewer #1 (Remarks to the Author):

The authors have addressed the raised points and I can recommend the manuscript for publication.

We thank Reviewer #1 for his very positive judgement.

Reviewer #2 (Remarks to the Author):

The authors now included language in the revised manuscript that address my main concerns. I am now supportive of publication of this manuscript.

We thank Reviewer #2 for supporting publication of this manuscript.